# Integrity, Trustworthiness, and Effectiveness: Towards an Ethos for Forensic Genetics

**DOI:** 10.3390/genes13081453

**Published:** 2022-08-16

**Authors:** Matthias Wienroth, Aaron Opoku Amankwaa, Carole McCartney

**Affiliations:** 1Centre for Crime and Policing, Northumbria University, Newcastle upon Tyne NE1 8ST, UK; 2Department of Applied Sciences, Northumbria University, Newcastle upon Tyne NE1 8ST, UK; 3Leicester Law School, Leicester University, Leicester LE1 7RH, UK

**Keywords:** effectiveness, epistemic culture, ethos, forensic genetics, integrity, justice, legitimacy, purpose, trust, trustworthiness

## Abstract

Forensic genetics comes under critical scrutiny when developments challenge previously accepted legal, ethical, social, and other boundaries. Forensic geneticists continue to build a knowledge culture within a community of practice that acknowledges ethical standards of conduct in both research and the societal application of forensic genetics. As the community further cements and extends its societal role, and in that process often pushing at ethical and legal boundaries, it requires a strong, resilient, and responsive ethos that, in setting clear parameters for conduct, fosters the field’s sense of purpose. While supra-national declarations and human rights protections, coupled with local regulations, provide some parameters for practice, and discipline-specific guidance has refined an agenda for forensic genetics research and application, this maturing field needs to now define its core principles. This contribution proposes the values of integrity, trustworthiness, and effectiveness as a foundational triptych for a bespoke forensic genetics ethos to ensure the augmentation of developments that range from a purely science-oriented to a wider societally relevant knowledge culture.

## 1. Introduction

Forensic science is considered not only a “*mainstay of the criminal justice system*” [1] but also “*essential to international security as well as global justice systems […] and is becoming increasingly important in the domain of human rights*” [2]. However, a widely recognised prerequisite is that forensic science must be “*grounded in ethical integrity, both in relation to scientific conduct and reporting*” [2]. The postulated essential and growing role of forensic science is thus not legitimate unless underpinned by rigorous ethical conduct in both research and case work. However, the sub-discipline of forensic genetics not only sits at and operates across the interface of basic (genetics) and applied science (forensics) with competing demands, norms, and values between science, law, policing, etc., but it is also a diverse field with stakes and community members from basic genetics, clinical research, forensic applications, case work, academic and policing labs. While the forensic genetics community may share views of issues and ways of knowing—of approaching, seeing, and interpreting problems—its practices can vary considerably between members. Arguments about the usefulness and importance of research and development in the field can vary between scientific, clinical, policing, or legal ones. Yet it is the social relationships between individuals and institutions from within the community and with those it interacts through case work, advocacy, and policy debates that drive the community’s development.

Forensic geneticists work within this broader cross-disciplinary field, utilising genetic science for legal purposes, most often associated with criminal justice, albeit this sub-discipline also continues to expand into areas such as disaster response and border management, among others. Recent scientific innovations coupled with this broadening of application, both pushing at previously negotiated legal and ethical boundaries, requires a strong *ethos* for forensic genetics: a sense of *spirit* drawing from a set of habits, values, norms, and relationships that inform the practice of its community. For the forensic genetics community to foster a strong knowledge and service programme against the background of competing commitments, obligations, and priorities, we propose an ethos of forensic genetics that spans ways of knowing and ways of relating, based on a foundational triptych of integrity, trustworthiness, and effectiveness. Integrity here relates to legal, moral, and ethical standing based on principled practice; trustworthiness is a potential (desirable and, arguably, necessary) product of integrity that impacts on the social standing of a community and its actions, what it can and cannot do; and effectiveness refers to evidencing claims of technoscientific capacities and capabilities of forensic methods, influencing what societal arguments can be made for forensic genetics. While most of what we discuss in this paper applies to the forensic genetics community per se, the one group most keenly addressed is that of scientists working on both research and case work.

Forensic geneticists span a complex web of organisations, ranging across a variety of social relationships that, like all human practices, are subject to an ‘order’ that will include explicit and implicit views on good conduct [3]. Sites of ethical decision-making may not always be apparent and ethical boundaries largely remain opaque, unregulated, and regarded as justifiably flexible under specific circumstances. Indeed, forensic geneticists operate in contexts where such justifications are readily located within discourses of ‘public interest’ and social utility, situated along a continuum of historically accepted practices of State collection and retention of personal biometric data in large collections (e.g., fingerprints and photographs). Reason perhaps that it is only belatedly that forensic genetics applications became subject to specific ethical scrutiny, arguably decades after case work began in the 1980s [4]. 

Recently, closer examination of ethical practice, notably around informed consent and data used for the development of and in forensic case work applications (including training sets and reference databases), has become necessary [5,6] after inadequate processes were detected in basic obligations such as reporting of ethical approval [7]. This follows closely on the heels of ethical debates surrounding new(er) applications around phenotyping, ancestry testing, epigenetics, and genetic genealogy. While not focussing here on these ethical issues (see instead [3]), we take such vital ethical considerations further and propose the foundations of an ethos that could underpin forensic genetics, ensuring its future as it advances and matures. 

## 2. Context: Human Rights and Legal Principles

Genetic data have long been afforded special protection and accorded particular significance in international declarations, many of which have given rise to rights enshrined in legal principles, including: respect for dignity and rights regardless of genetic characteristics (Art 2) [8]; non-discrimination and non-stigmatization of individuals, families, groups, or communities on the basis of DNA (Art 7) [9]; and the prohibition of drawing direct linkages between an individual’s DNA composition and their ethnicity or nationality [10,11]. For forensic genetics, these principles are significant because results of DNA testing may lead to discriminatory profiling based on ethnicity, gender, race, or other (supposedly genetic) characteristics [12,13,14,15]. Furthermore, the UN Human Rights Council’s Resolution on Forensic Genetics and Human Rights urges States that forensic genetics should be undertaken in compliance with accepted international scientific standards, including the utmost respect for confidentiality [16]. This matters particularly for the types of analyses undertaken in forensic DNA phenotyping and biogeographic ancestry testing, forensic genetic genealogy, and forensic epigenetics (as their group-focus opens up new ethical questions); and unethical conduct in research can quickly lead to questions of integrity and mistrust when applying these technologies in case work and other domains of deployment (think ‘fruit of the poisonous tree’).

Principles of personal autonomy, human dignity and human rights are echoed across various declarations issued by the Council of Europe on DNA analysis in the criminal justice system [17], biotechnologies [18], the protection of the human genome [19], and the human rights considerations of biometrics [20]. Most often however, debate over forensic genetics centres upon the right to privacy: in *S & Marper vs. UK* [2008], the European Court of Human Rights (ECtHR) confirmed that the ‘protection of personal data’ came under Article 8 rights to privacy [21] and European jurisprudence makes it clear that the retention of human cellular material is particularly sensitive. In *van der Velden v. the Netherlands* [2006], the taking of bodily samples for DNA testing amounted to an intrusion on the applicant’s privacy given the use to which cellular material could conceivably be put in the future. The unanimous ruling in *S & Marper vs. UK* added: “*In addition to the highly personal nature of cellular samples, the Court notes that they contain sensitive information about an individual, including information about his or her health. Moreover, samples contain a unique genetic code of great relevance to both the individual and his relatives*.” (paragraph 72). The retention of cellular samples is “*particularly intrusive given the wealth of genetic and health information contained therein*” (paragraph 120). In *Aycaguer v. France* [2017], the ECtHR reaffirmed that this relates not just to cellular samples, but to the profiles generated, which, although they contain more limited information, can go beyond ‘neutral identification’. 

While genetic information is thus afforded particular attention in declarations, and legal protection via privacy rights considered ‘fundamental’ (i.e., other rights are dependent upon it), even an expansive respect of human rights and an avowed adherence to such declarations may not ensure that forensic genetics remains beneficent. An ethos underpinning forensic genetics may be the next step, therefore, in securing the future of the discipline, built on a synergy of scientific and social justice commitments.

## 3. Why an Ethos for Forensic Genetics?

A relatively strong international framework of human rights protections is thus in place when it comes to the retention of genetic material for forensic purposes, albeit with a limited focus on privacy and databases. National legislation governing forensic genetic databases is common, but not universal, and most often does not encompass all issues relevant to the processing and use of biometric samples and data. Far less common is national legislation addressing the use of more advanced genetics analyses, which tend to be arcane and in flux, neither providing a strong reflection of ethical values nor a consideration of the effects of recent and emerging biometric data types (including population-focused and genomic ones) and their uses. Neither national nor supranational legal frameworks can incorporate all principles contained within the raft of relevant international resolutions and agreements: *legal governance is limited and finite*, and is not directed at ensuring the ethical conduct of the forensic genetics community.

There are relatively clear ethical frameworks governing genetics and genomics research in biomedicine that emphasise the responsibility of the researcher to research participants and to society. Strict adherence to them is vital in order to avoid public and policy backlashes, such as those seen over unethical human genome editing experiments in China and Mexico [22,23], and non-consensual collection of DNA materials and data from minorities in China and Europe [24,25]. While this shows that ethical conduct within biomedical genetics research can also still be erratic, the data collection from minorities and marginalised communities, in particular, has recently required focussed attention within forensic genetics research. Continued scrutiny is thus equally required at the juncture of forensic and biomedical genetics, when medically collected or relevant knowledge and data may be used in forensic contexts. 

In many countries, there have been, over the last decades, a series of government and non-governmental reviews, inquiries, reports, and post-mortems of wrongful convictions, critical of the role played by forensic science [26]. Many have prompted efforts to improve or ‘regulate’ forensic science, such as the creation of oversight bodies, e.g., the UK Forensic Science Regulator (FSR) [27]. The FSR, in collaboration with the forensic science community, produces regularly updated ‘Codes of Practice and Conduct’ (https://www.gov.uk/government/collections/forensic-science-providers-codes-of-practice-and-conduct, accessed on 19 July 2022), setting out standards and norms of practice to be adhered to by all forensic science practitioners. The forensic science community has professional bodies, with regional Forensic Science ‘Academies’ or ‘Societies’, with sub-committees or groups focussed upon genetics. For example, the European Network of Forensic Science Institutes (ENFSI, https://enfsi.eu/, accessed on 19 July 2022) was created in 1995 to improve the quality of forensic science in Europe, including an Expert Working Group on DNA. Most recently, the eponymous ‘Sydney Declaration’ [1] called for the discipline to return to the ‘essence’ of forensic science, with a recommitment to fundamental principles to solidify a robust scientific basis for the field. 

Most often in such reports and responses, forensic genetics has been cast as the ‘gold standard’ for other forensic disciplines to emulate, and due to its (perceived) greater scientific ‘rigour’ than many other forensic disciplines, has avoided censure. Aside from the strong scientific basis for forensic genetics, this eminent position may also be attributable to earnest efforts since the emergence of the field, to ensure uniform high standards. The International Society for Forensic Genetics (ISFG, https://www.isfg.org/, accessed on 19 July 2022), founded in 1968, with members from over 60 countries, promotes scientific advancement in the discipline and provides scientific recommendations and advice on best practice. This role was further enhanced in 1988 with the creation of the European DNA Profiling group (EDNAP, https://www.isfg.org/EDNAP, accessed on 19 July 2022), seeking to harmonise DNA profiling technologies for crime investigations. In the UK, in 2018 (and updated in 2020), the Biometrics and Forensics Ethics Group (https://www.gov.uk/government/organisations/biometrics-and-forensics-ethics-group, accessed on 19 July 2022) developed and published a set of high-level governing principles that should be applied to the development and use of biometric and forensic technologies.

The forensic genetics discipline has thus grown and matured into a community of practice; regularly interacting professionals who share concerns, interests, and practices (for the concept of ‘community of practice’ see [28,29]). This community articulates an epistemic culture of forensic genetics-defined practices of how knowledge is produced and validated, which may at times imperfectly align with commercial aims for the exploitation of forensic technologies [30]. In addition, while some questions of professional ethics may be currently addressed in respect of research, the same cannot be said for case work, which, if governed at all, is mostly done via tenuous legal regimes and voluntary codes of conduct [31]. Forensic geneticists thus must continue to individually navigate divergences between science and (criminal) justice, as well as expertise and non-expertise, [32], engage in ethical boundary-work [33,34], and continue in their efforts to proactively build a community [35]. 

Some have called for the discipline to refocus its purpose [36]. While agreeing that the question of purpose is fundamental, we emphasise the creation of a self-governing community of practice (with a shared vision of purpose), with a collective commitment to a bespoke ‘ethos’ for forensic genetics while subscribing to a robust ethical scientific basis for forensic work—as the Sydney Declaration has, in part, suggested. A strong ethos will support the development of and adherence to professional ethics in both research and application, cementing the building of a trustworthy community and epistemic culture. This includes—through reflection on, and development of shared values, norms, and good practice—building resilience and responsiveness to both challenges to forensic genetics’ standing and to hyperbolic claims around the capacity of DNA to facilitate security and justice. This means that validity, utility, and legitimacy (all questions of integrity, and eventually of trustworthiness) [37,38] are as essential to the field as scientific reliability. 

While the forensic imagination has to date been focused on the power of DNA to provide forensic intelligence in police investigations and evidence in criminal trials, this imagination is expanding to incorporate ‘justice’ in (global) society more broadly (e.g., organised crime prevention and deterrence; anti-terrorism; public safety and health; family reunification; disaster victim identification; border control; war crimes; missing persons; etc.) [26]. Such ambitions require an ethos of forensic genetics that can span a far larger web of local, regional, and international organisations and more complex and contentious domains. While legal parameters have to date been considered sufficient by police and domestic criminal courts, as the reach and impact of forensic genetics expands, so too must governance and oversight. The advancement of forensic genetics as a distinct field requires the concomitant maturation of an overarching sense of purpose and societal standing, with greater sophistication, ambition and scope, with integrity, trustworthiness, and effectiveness as core values.

Simon Cole in 2013 wrote “that the solution posed by mainstream scientific institutions like the NAS [US National Academy of Science]—that forensic science ‘adopt[] scientific culture’—while perhaps a noble idea, is unrealistic. It is unrealistic not merely for the oft-stated reason that forensic scientists and those who employ them have evinced resistance toward such goals. More importantly, the social structure of forensic science is fundamentally different from that of research science” [39]. He argues that there is a very particular epistemic culture in forensic science that adventitiously—rather than intentionally—produces data, where volume and speed of analysis as well as unambiguous reporting of data are highly priced. For forensic genetics, the situation is somewhat more complex as basic science has a vital role in the development of new methods, and academic scientists are central to the forensic genetics community of practice. However, the point stands that it would be very difficult to ‘enforce’ a singularly scientific culture on forensic genetics when its constituents are so diverse and include non-scientists. Nonetheless, with Cole, we argue that there is a shared epistemic culture that can be the basis of an ethos for forensic genetics. With ethos, we refer to a sense of spirit or character of the community. The ethos frames the community’s epistemic culture—of how knowledge is produced and organised—and reflects the aspirational sense of a community’s locus in society [40].

## 4. Integrity

‘Integrity’ is an easily grasped concept that can refer to both physical condition (e.g., being ‘whole’ or ‘stable’), as well as moral and ethical standing (e.g., being ‘honourable’ or ‘principled’). While there are a variety of international statements on research integrity (e.g., [41,42]), the demands of integrity on forensic genetics’ practical applications and case work exceed those in such statements. The use of forensic genetics in the criminal justice process does not diminish the need for procedural justice; in fact, its success should always be gauged by how well it secures and maintains fairness of such processes. Such fairness should always be demonstrable, and the use of technology and data must be transparent, since a justice system “must assess itself not only against narrow criteria of crime control, but against broader criteria relating to people’s trust in justice and their sense of security” [43]. 

As a multi-faceted concept, attempting to cover the diversity within the community of practice, integrity in forensic genetics should be assessed against a matrix of standards to be met or achieved (see Figure 1). There are clear commonalities with criteria present in regulatory models, ‘Codes of Practice,’ and guidelines that proliferate around not just DNA, but other biometric data and technologies such as AI. Criteria for (anticipatory) governance (e.g., [35,44]), as well as terms of reference for oversight bodies, include many of the same or similar benchmarks. When combined, such criteria provide a holistic evaluation of the ‘integrity’ of forensic genetics. 

The viability of the discipline is assured by guarantees that the forensic genetics techniques are valid: they ‘work’ and can be operationalised (although effectiveness, while central to integrity, requires specific focus due to its vital part in helping to legitimise the role for forensic genetics in society). Data gathered and produced must be reliable and universally understandable without complex translation or interpretation that could lead to confusion and variability (e.g., efforts have been made around evaluative reporting [45,46]). Systems and processes should be guaranteed (as far as possible) by robust quality assurance mechanisms aligned with internationally agreed standards. 

Importantly, legitimacy is about human rights compliance. As this can be difficult to navigate in practice, forensic genetics research and application must be lawful, and no work should be undertaken (without detection and sanctions) outside of the law. Both research and application take place within enforceable ‘boundaries’ with no data used for non-permitted purposes or shared outside of lawful permissions, preventing abuse of data and ‘mission creep’. At the same time, legal boundaries are only temporarily settled agreements and do change when new understandings are perceived as more legitimate than existing ones and lawmakers are swayed to revise or implement new legislation (e.g., see forensic genetics-relevant changes in the German Code of Criminal Procedure in 2019; frequent changes in Dutch phenotyping legislation; ongoing Swiss debates around the legislation of advanced genetic profiling techniques; etc.).

Therefore, critically, forensic genetics research and applications must also be socially acceptable. It needs to be appropriate and justifiable, *across time and place*, judged not only by the forensic genetics community and partners, but also by independent bodies making the community accountable and transparent [27,47]. Most obviously, key ethical and legal principles, embedded within a governance structure and based on widely shared ethical values reflective of pluralistic public expectations, must be respected. If forensic genetics does not have broad public support, then enthusiasm will wane, and it will be very difficult to regain or retain confidence and credibility (and funding may diminish in line with weak political motivation). There must be sufficient information for the public and policy makers to assess the (cost) effectiveness of techniques, to assess the contribution to public security. Most obviously, a robust research programme should assess the *end-to-end probative value* of both DNA databases and advanced forensic DNA techniques. There can then be proper consideration of the costs and benefits of forensic genetics in order to make evidence-based decisions on their parameters.

Lastly, but perhaps most importantly, is the role of oversight bodies with adequate capacity and comprehensive powers to enable meaningful scrutiny. To ensure integrity, oversight requires evaluation of scientific and operational validity as well as demanding proof of adherence to legal and ethical requirements, coupled with continuous monitoring of efficiency and acceptability. Regulatory structures should also be capable of anticipating as well as responding to issues and would include: dispersed responsibilities across multiple agencies incorporating diverse perspectives; transparent policies and decision-making criteria; accountability and compliance mechanisms; ongoing evaluation, and public/political dialogue. Actors and bodies tasked with oversight should be enabled to conduct research and derive rules/guidance. Wherever possible, these should be statutory bodies with sufficient resourcing and powers to be effective. The forensic genetics community has a key role to play here.

## 5. Trustworthiness and Trust

Without scientific conduct being perceived of as trustworthy, science’s power of making rigorous and reliable statements about the natural world and their impact on the social world are diminished in the eyes of those using, and being subject to, scientific and technological interventions in society. Arguments of whether science ought to be trusted also exist for forensic science (e.g., around forensic hair or ear print analyses), and such debates are vital in negotiating the parameters of trustworthy science. Consider the contestation of forensic genetics methods in US-American and British courts from the late 1980s to the early 2000s. Courts’ views of the trustworthiness of evidence from new and emerging forensic genetics analyses and their experts rested in good part (and still do) on the ability of forensic geneticists to communicate evidence persuasively, but also by making necessary changes, including harmonisation and validation of methods and technological processes such as DNA extraction, analysis, and communication of findings. This responsiveness—to act and evidence action—has contributed a level of *trustworthiness* for forensic genetics in criminal justice. Similarly, actions—or inaction—can undermine trustworthiness. If users and publics of forensic genetics were to find the underlying science, one of its technologies, or the experts, as untrustworthy, the community would not only lose funding, but its services and standing would be discredited to the effect that its analyses would be highly contestable, users may avoid procuring services because the produced intelligence and evidence would be considered untrustworthy, and legislation may become more stringent.

Whereas integrity is a collective effort from within the forensic genetics community in collaboration with partners and stakeholders, trustworthiness emerges from integrity being recognised by those with whom the community engages. Importantly, forensic genetics as a field cannot demand or generate trust. Once someone or something is considered trustworthy, however, trust can be generated. Scientifically and socially robust forensic genetics practices and ethical conduct can provide points of reference for *trustworthiness*, but it is important to understand that others need to be able to recognise someone worthy of trust in order to *engender trust* in the system(s) in which they operate. Having said that, different points in a process, or elements of a system can be deemed to be more or less worthy of trust than others: trustworthiness is *situational*. This links to trust relationships being based on expectations about future actions, on predictability and mutuality in situations of lack or an imbalance of knowledge [48]. Trustworthiness is *relational*. The trustworthiness of forensic genetics also plays a particularly interesting societal role because not only is scientific conduct subjected to questions of trustworthiness here, but also those elements that draw on forensic genetics analyses in order to make decisions around, e.g., suspicion and culpability in the criminal justice system. 

Trustworthiness of forensic genetics is essential to the maintenance of confidence in policing and the criminal justice system (and increasingly other domains), critical for ensuring public acceptance and compliance with findings of forensic genetics work. These domains (justice, national security, public safety, etc.) are central to important social and ethical outcomes. Forensic technologies, in particular innovative techniques, and the police powers required to utilise them, influence, and are influenced by, the development of social order (e.g., legal reforms, policing practices, etc.). The forensic genetics community of practice needs to bear the responsibility of their influential role in the social order with the requisite solemnity. 

Trustworthiness of forensic genetics fluctuates, e.g., when there has been a reported failing or controversy: “*cases where a major miscarriage of justice was caused by an erroneous DNA result often generate a lot of media attention and damage the reputation of forensic laboratories*” [49]. Vocal concern is viewed as a fundamental threat to the community and has sometimes provoked defensive reactions. While trust tends not to be qualified, let alone quantified, it is often cited as the quintessential prerequisite for forensic genetics. After all, the forensic genetics community tends to claim to be providing evidence of unrivalled power where “*the allegedly high level of sophistication and complexity […] is commonly thought to be inaccessible to non-experts*” [50]. Here, but also more widely, the centrality of trustworthiness in science as in forensic genetics is grounded in the realisation that “*most citizens have little alternative but to put their trust in what they can judge about scientific practice and standards, rather than in personal familiarity with the evidence*” [51]. Reflective practice and responsible communication of how and why an analysis is appropriate, where its limitations lie, or recognition that something has not worked well, can contribute to building trustworthiness.

There has been a long history of contestation of forensic genetics in policing and the law [52,53,54] and forensic genetics continues to simultaneously attract both high levels of confidence and concern. This is reminiscent of debates around the public perception of science more generally [55,56], especially when it comes to communicating and regulating risks of scientific endeavours [57,58]. A key point is that viewing trust as a means of enabling certain practices and decisions negates the very idea of trust [59]. Trust in the scientific basis of forensic genetics is raised time and again, but other aspects merit attention: e.g., the marshalling of forensic information by non-scientific stakeholders in different operational contexts, and the fitness and contribution to decision-making in a variety of spheres (e.g., criminal justice, border management, disaster response, etc.). Trustworthiness is *dispersed* across different sites and practices, which forms part of its relational character. Forensic genetics features at the intersections of multiple influential societal domains—most obviously science, law, policing (arguably also medicine)—and is subject to competing interests and priorities leading to specific rules, practice, and standards that shape invocations of trust. In criminal justice alone, diverse non-scientific stakeholders have their expectations set by forensic geneticists and work with information provided by them. These range from law enforcement agencies, prosecution, and the judiciary, to juries, victims, defendants, their families, and beyond to communities of minorities and the citizenry. Yet what one of these parties may see as sufficiently trustworthy (e.g., based on scientific principles, forensically validated, subject to professional standards, lawful, etc.) may be considered insufficient by those with additional demands, such as the compatibility of professional practices with procedural justice, human rights compliance; demonstrable effectiveness and efficiency; or a discernible impact on public security. 

Concurrently, legal and policing practices must also be deemed trustworthy by scientists involved in case work. Where policing must be undertaken with citizens’ consent and accountability, trustworthiness is reflected through public and political consensus. After all, trust is reciprocal and *relational* [60] and invocations of trust are often made specific to interpersonal relationships. However, when it comes to specialised domains such as science (including forensic genetics), the (inter)personal is superseded by *institution-level trust relationships* at the interfaces of science, law, and policing. Trust here fulfils a *stabilising function for a system* [61]. Such “*systemic trust*” is a fragile achievement constituted in part by codified practices and the building of a community of practice contributing to the overall integrity of a system [62]. In the forensic genetics community focussed on the criminal justice system, the loci of trust negotiations span epistemic (scientific), operational (policing), and courtroom (legal) practices: at all three instances, the integrity of forensic information is negotiated and tested [62]. Other sites of trust negotiation as well as mutual defining what renders something and someone trustworthy are relevant when forensic genetics operates outside criminal justice. 

## 6. Effectiveness

While effectiveness is essential to integrity, we emphasise its central role in a strong ethos for a forensic genetics community. Considering the special significance accorded genetic data, it is critical that the powers afforded to States to seize, process, and retain such sensitive data, infringing upon individual bodily integrity and autonomy, and diminishing privacy rights, has powerful justification. Effectiveness plays a critical role in equations of ‘balance’ when determining the viability, legitimacy, and acceptability of forensic genetics. It is vital, therefore, that attention is paid to defining the public ‘goods’ that are to be achieved, and then assessing whether forensic genetics is actually achieving these, in whichever domain they are applied. To argue that something is effective, it must then be demonstrated that actual outcome(s) meet predetermined goals, standards or expectations. An important interrelated demand is that it is also ‘efficient’: the cost of the achievement of aims (e.g., crime detection) using forensic genetics tools is favourably compared to that of alternative systems (e.g., employing more detectives). As such a cost/input-benefit analysis is required, albeit the actual ‘costs’ of forensic genetics are rarely measured and are very narrowly conceived [63]. 

In criminal investigations, outcomes can be complex, evidenced by the multiple metrics proposed to assess effectiveness [64,65]. Different applications may also have different aims, ranging from identification (and ultimately the ‘matching’ of DNA profiles), to providing intelligence (indicating possible kinship, or inferring appearance or ancestry, for example). In England and Wales, the national DNA database is lawfully established to: (1) protect national security; (2) assist terrorist investigations; (3) assist in the prevention, detection, investigation, and prosecution of crime; (4) assist in identification of a deceased person or verification of identity (*Police and Criminal Evidence Act 1984* (PACE) s63T(1)). These outcomes are clearly idealistic, given that at its height, DNA evidence can only suggest the possible presence of a DNA profile that matches an individual, found in an incriminating location. It ordinarily can indicate little or nothing about the activity of a person, nor when the DNA was deposited, whether they were actually present, or how their DNA may have arrived where it was located. If admitting to such qualifications, convictions based upon DNA evidence require very careful deliberation, not always undertaken reliably by the Courts [66]. For DNA ‘intelligence’, there are yet more caveats, while for some ‘outcomes’, such as the prevention of crime or protection of national security, arguably, there can be no supportive evidence of effectiveness. 

Various measures have been proposed to evidence effectiveness, each typically reflecting very narrow parameters of assessment [63,64,65]. Evaluations are also necessarily dependent upon prior expectations, which as explained, mostly remain idealistic, ill-defined, and omitting complementary accounts of broader impacts, which are themselves similarly vague and disputed. In attempting to assess perhaps the most obvious and straightforward of aims—the detection and prosecution of crimes—measures of the impact of forensic genetics use different data, utilising different calculations, and unsurprisingly reach varying conclusions. Existing studies often miss important context about the actual use of forensic genetics in investigations and prosecutions, and so understanding of the factors limiting the effectiveness of forensic genetics remains equivocal. 

This knowledge gap most often leads to overblown claims about both the current benefits and future capabilities of forensic genetics. Indeed, the forensic genetics literature, as well as commercial publications and outputs, have faced criticism for over-selling techniques and technologies, while under-playing limitations or poor results. Examples are plentiful where the ‘hype’ surrounding a technique (e.g., familial searching, ‘Rapid’ DNA, DNA ‘photo-fits’, biogeographic ancestry testing) has not been matched by results. Scientific papers demonstrating the potential of techniques within laboratory test environments ought to be supplemented with details of their application to real-world investigations, demonstrating the effective (and ethical) translation of laboratory results to police investigations. It is often highly speculative and requires extreme caution when extrapolating from laboratory-based experimentation to the realities of real-world application. The adoption of a technique into police investigations requires a great deal of consideration and care. The greater the incursions into privacy, the greater the justification required. If advanced forensic genetics techniques pushing at the boundaries of ethical practice are not particularly effective, then, arguably, they cannot be justified. While greater efforts may be required to improve their effectiveness, concomitant efforts must also be put into reliably demonstrating effectiveness. If effectiveness can be proven, then this may perhaps also help to bolster trustworthiness of forensic methods and public acceptability, while a lack of trustworthiness will be generated by hyperbolic claims that ultimately are not substantiated. 

## 7. Discussion

The totemic status of DNA and particular sensitivities of genetic data are recognised by a variety of international declarations, and yet national legal regimes and supra-national legal rulings offer weak parameters that remain malleable. National and international debates surrounding forensic genetics universally conclude that the field requires careful regulation, with mature and effective oversight and robust accountability mechanisms. Yet these are most often either absent, only partially effective, or beset with questions of independence, partiality or commercial pressures. The forensic genetics community, having matured into a professional community of practice, is now more favourably positioned than ever to develop a bespoke ethos, to future-proof the discipline and augment efforts already undertaken to guarantee ethical practice and scientific rigour. 

A foundational triptych for such an ethos includes the concepts of integrity, trustworthiness, and effectiveness. Integrity is vital for the viability of a community of practice, for self-confidence and morale within the community, but also for stakeholders and publics to find the forensic genetics community worthy of trust. To ensure the integrity of forensic genetics, critical attention must be paid not only to the viability of the science and its application, but also to its legitimacy and acceptability, including the ethical erasure of material/records [67]. The ‘integrity’ of forensic genetics research and application must encompass all work undertaken by forensic geneticists, including efforts to extend their remit, and influence law and policy. Integrity engenders trustworthiness, and therefore is essential to forensic genetics, its community, judicial bodies, law enforcement agencies, and justice systems more broadly. Trustworthiness constitutes good ethical practice and trustworthy agents can engender trust. Trust is not a gift but a (temporary) representation of mutual expectations within a system. Trust in forensic genetics, its practices, ethics, and practitioners, is a vital *stabilising* element of each of the systems to which forensic genetics contributes (justice, national security, public safety, etc.). 

Yet trustworthiness and trust can only be present when there is both an assurance of integrity, and a measure of effectiveness: it can be relied upon to work. The authority and reliability of claims made about forensic genetics remain obscure at best. While we know forensic DNA analysis can be powerful in individual cases, its contribution to criminal detection is mostly undetermined. Whatever the confounding variables that may be impacting upon the ability of DNA databases, in particular to improve detection rates, the available evidence indicates that their aggregate contribution to the resolution of crime remains stubbornly low [68]. The future of the field and its impact on security and justice will depend on identifying specific areas where genetic analyses are most useful, to focus resources, and prevent ill-advised expansion into areas where it cannot be justified by positive outcomes. If unable to demonstrate how useful forensic genetics is, then it is difficult to justify the intrusions into privacy, personal dignity, and bodily integrity. Are the means justified by the ends?

Valid and reliable evaluation should therefore be a requirement of any forensic genetics regime, and thus, the forensic genetics community should be highly motivated to make vigorous efforts to demonstrate effectiveness. Such measures of efficiency and effectiveness must be integrated with broader considerations to achieve both a realistic and holistic view of technology ‘utility’ [69]. Proper consideration of costs and benefits are essential to make evidence-based decisions on parameters. The ethical and financial costs require that technologies develop, and are implemented progressively, with decisions regarding the constitution of DNA regimes based upon realistic evidence, rather than pursuing an expansionist agenda based upon over-inflated perceptions of benefits that could be accrued. Such arguments can be extrapolated to the introduction of new processing/analysis tools. 

Of course, it is notoriously difficult, if not impossible, to find the ‘optimal’ (acceptable) scale and arrangements for DNA profiling and testing, when you cannot ‘weigh’ matters such as the public benefit derived, or the detriment to rights. Reviewing so-called public interests versus private (but also communal interests where the rights of (minority) groups may be disadvantaged in favour of an overly generic ‘public interest’), requires consideration of the ‘necessity’ of DNA profiling and retention, what data are relevant for achieving predetermined aims, and how to approach forensic testing and its communication to investigators and the judiciary as well as publics (e.g., local communities, victim groups, civil liberties organisations, policy-makers). There must then be a calibrated gauge which sets the effectiveness of forensic DNA profiling (the public benefit) against any negative consequences. However, the notion that any such gauge can be calibrated is fanciful when dealing with amorphous constructs and variables that are not (cannot?) be measured. Any gauge may not even reflect true relationships between constructs such as ‘security’ and ‘privacy’, if based upon an assumption that any extension of individual privacy rights (which are also public privacy interests) compromises ‘safety’ on the other side of the metaphorical scales. Experts explain that this is not how the safety/privacy equation works: indeed, increasing privacy can create a more secure society (more privacy = more security) [70]. Targeting individuals (or groups) in society and infringing their rights for the purported benefit of others, leads to neither security nor justice.

In addition to idealistic outcomes such as ‘crime resolution’, variations of which are commonly found in enabling legislation across countries, the European Court of Human Rights, as well as stakeholders call for additional goals including broader civil liberty aims [21,71,72]. Such aspirations are rarely articulated (while they may be referred to when extolling the virtues of forensic genetics they are not committed to), but Bieber [65] details the importance of personal and societal interests in forensic genetics, because of the *reciprocal relationship* between the public and the State (and thus law enforcement): the social contract between citizen and state transfers the monopoly of force (e.g., in criminal investigations, public safety, etc.) to the state in return for security (also from the state) for the citizen. In democracies policed by consent, citizens must accede to the powers necessary to lawfully enable forensic genetics, thus shaping legislation and policy governing forensic genetics. Citizens should then benefit from permitting such powers, and then cooperate with their lawful implementation. An effective forensic genetics application should therefore achieve social justice outcomes, including bolstering the civil liberties and human rights of individuals and communities. If the forensic genetics community’s aim is to contribute to security and justice, then the community should be yet more ambitious (within the bounds of integrity and legitimacy) and make good on the rich promise of forensic genetics.

## 8. Concluding Remarks

We humbly offer one pathway to answering the question of how forensic genetics can mature in a resilient and responsible manner, believing that the time has come for the forensic genetics community to adopt a bespoke ethos, a sense of spirit built on ethical practice (integrity), on a relational character (trustworthiness) that recognises its responsibility to society, and the need to scientifically and operationally evidence impact (effectiveness). As this innovative field continues to grow in influence, increasing applications for its considerable repertoire of knowledge and technologies, it ought now to define a resilient and responsive ethos, providing a steadfast path along which the community should travel. This should (re)affirm strong societal commitments, especially in light of ambitions to broaden the role of forensic genetics in support of social justice and human rights beyond criminal justice. In developing a strong ethos, we need to be mindful that, as the then UK Government Chief Scientific Advisor Mark Walport wrote in 2014, “*We can only have the best discussion about innovations if we understand that the discussion must be about both science and values*” (p. 7) [73]. This does not mean to confuse social values with science, but to accept and engage in discussions about how scientific insights and technologies can be developed and used according to shared ethical values. Such discussion starts at the point of scientific values: e.g., how are samples and data collected; are the chosen data representative, are they reliable, what are their biases and how can they be mitigated; to what purposes is something researched or reported in forensic genetics? Here, values about robust science, rigour, effectiveness, and integrity play a vital role in the negotiation of who and what are trustworthy, and under what circumstances. The three core values of integrity, trustworthiness, and effectiveness thus correlate with the key qualities of forensic geneticists, as arbiters of the interface of science, justice, and social responsibility.

Such an ethos can thus only serve to strengthen and further grow this community of practice, to ensure the future benefits from their scientific endeavours. Recent work towards developing a core for forensic genetics [1,36] has focused on robust science, disciplinary concerns, and on the criminal justice system. A strong ethos would support the discipline in realising future ambitions. If science is the brain and casework the heart of forensic genetics, then its ethos is its soul, bringing both together with a strong sense of purpose and spirit.

## Figures and Tables

**Figure 1 genes-13-01453-f001:**
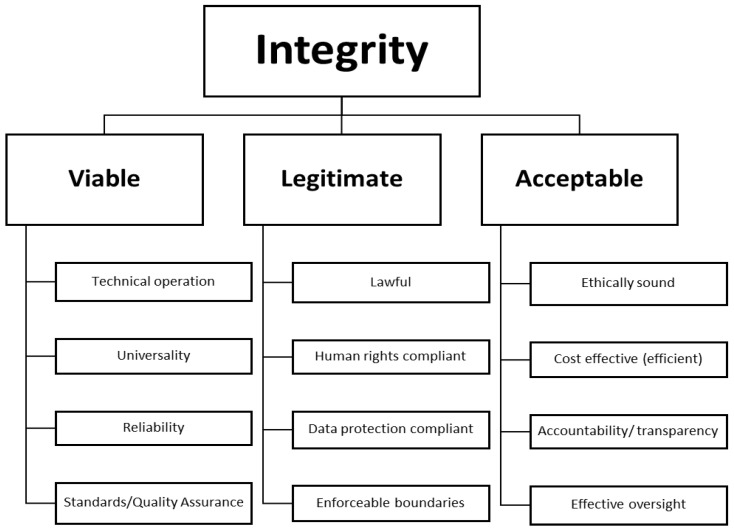
The ‘Integrity Matrix’.

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
