# Peer review of "Integrity, Trustworthiness, and Effectiveness: Towards an Ethos for Forensic Genetics"

_genes, 2022, doi:10.3390/genes13081453_

Round 1

Reviewer 1 Report

This paper systematizes important ethical and social values related aspects to take into account in the current state of transformation of forensic genetics, marked not only by the growing use of techniques to analyze the human genome, but also by the fact that forensic genetics operates in a societal context marked by growing international cooperation; of very different social actors - from basic scientists,  to laboratories working for courts to police forces.

I am not sure if the word ethos (derived from the Greek word for custom, habit, or usage) is the best option for defining the core argument the authors are willing to convey. In my understanding, the authors are referring to ethical standards and social values. I also did not quite understand what is, in the authors’ views, the actual difference,  in the context of the present text, between ethos and epistemic culture.

I believe that the paper develops a  consensual arguments, which is why, despite having doubts about the original and innovative contribution of this text (this aspect does not seem to be explained), I recognize that the paper presents a robust systematization of the main aspects that should guide the conduct of the community of forensic geneticists. The definition of each of the principles can also be useful for dealing with more complex situations, por examples, ones which rise particularly challenging ethical challenges. However, I do have some comments and questions, which I will now briefly describe.

First, I believe that the authors do not clearly define the main aspects of diversity that characterize the field of forensic genetics. Simon Cole (2013) speaks of a forensic culture that is characterized by the blurring of boundaries between what is commonly called basic research and applied science. On the applied science side, what happens with forensic genetics is that it often 'only' responds to requests from the justice system and not to scientific peers. The hybrid and multifaceted character of forensic genetics is further reinforced by the diversity of actors and organizations that develop this type of knowledge and analysis: from scientists who work mainly in fundamental research, to laboratory technicians who respond to external requests and elements of police forces. From my point of view, the lack of consideration of this aspect weakens the argument developed in the text, as we do not know who are the main recipients of such values ​​to which the authors refer. Although we can admit that the answer will be something like "it's a proposal valid for everyone", even so, it would be worth discussing taking into account different epistemic cultures and readings of ethical principles that mark professionals from different areas of activity. Being a scientist is not the same as being a police officer.

In terms of the structure and organization of the text, it would be useful to the reader if the definition of each of the concepts - integrity, trust, and effectiveness - appeared immediately, in a very abbreviated format, at the beginning of the text, reserving the detailed description for the different sections that make up this article.

Finally, the authors' approach of 'trust' seems reductive to me, and I wonder what the real difference is in relation to other texts (an excellent example of this is the article “Value beyond utility: Let’s R.U.L.E.” (Wienroth, 2020; Journal of Responsible Innovation). I would like to see a more elaborated notion of trust, that also addresses “trustworthiness” (of individuals and institutions) and problematizes traditional accounts of “public trust”. An potentially useful reading might be  the text by Kieran O’ Doherty (2022), “Trust, trustworthiness, and relationships: ontological reflections on public trust in science” (Journal of Responsible Innovation)

I believe this article should be published, but with the additional clarifications and amendments I suggested.

Author Response

Dear Dr. Turchi, dear Ms. Long,

We are grateful to the reviewers’ time for commenting on our manuscript, and we appreciate the opportunity to respond to their comments. We are particularly delighted by Reviewer 2’s comments: “An excellent paper which will make an important contribution to the evolving field of forensic genetics. The paper is exceptionally well-written, referenced extensively and makes a solid contribution to the debate around use of this technology. […] A pleasure to read - look forward to seeing this paper in print!”

We have considered all comments carefully, please find our responses below. We have tracked all text changes in the attached revised manuscript for your ease of access (and we have also attached a clean file with all changes accepted). NB: We spotted a doubling of a reference (previously as endpoints 59 and 63) and made the necessary changes to the order of endpoint references (from endpoint 63 onwards) which we have not tracked in the revised manuscript. Since we are working with a document provided by the journal now, any new / added references are numbered from the last reference endpoint onwards, irrespective of their place in text.

All the best,

The authors.

Reviewer 1

R1: I am not sure if the word ethos (derived from the Greek word for custom, habit, or usage) is the best option for defining the core argument the authors are willing to convey. In my understanding, the authors are referring to ethical standards and social values. I also did not quite understand what is, in the authors’ views, the actual difference,  in the context of the present text, between ethos and epistemic culture.

  • Response: We agree that we can be clearer about what we mean by ethos and have provided a definition at the start of section 3.

I believe that the paper develops a  consensual arguments, which is why, despite having doubts about the original and innovative contribution of this text (this aspect does not seem to be explained), I recognize that the paper presents a robust systematization of the main aspects that should guide the conduct of the community of forensic geneticists. The definition of each of the principles can also be useful for dealing with more complex situations, por examples, ones which rise particularly challenging ethical challenges.

  • Response: We agree that our contribution to the discussion of purpose for forensic genetics provides a robust analysis. Our original contribution lies in offering one resolution pathway to the often posited but never resolved question of how forensic genetics can develop further, which we have emphasised again in various points throughout the paper. We draw from key work on ethical, legal, and social aspects of forensic genetics but go beyond such discussions in proposing the foundation for a maturing spirit of the community, the community’s character as a whole. By doing so, we also go add to and further extant forensic genetics contributions on the future of the field.

However, I do have some comments and questions, which I will now briefly describe. First, I believe that the authors do not clearly define the main aspects of diversity that characterize the field of forensic genetics. Simon Cole (2013) speaks of a forensic culture that is characterized by the blurring of boundaries between what is commonly called basic research and applied science. On the applied science side, what happens with forensic genetics is that it often 'only' responds to requests from the justice system and not to scientific peers. The hybrid and multifaceted character of forensic genetics is further reinforced by the diversity of actors and organizations that develop this type of knowledge and analysis: from scientists who work mainly in fundamental research, to laboratory technicians who respond to external requests and elements of police forces. From my point of view, the lack of consideration of this aspect weakens the argument developed in the text, as we do not know who are the main recipients of such values ​​to which the authors refer. Although we can admit that the answer will be something like "it's a proposal valid for everyone", even so, it would be worth discussing taking into account different epistemic cultures and readings of ethical principles that mark professionals from different areas of activity. Being a scientist is not the same as being a police officer.

  • Response: We refer to the boundary nature of forensic genetics throughout the text but recognise that this can be strengthened because it is such a vital element of considering the epistemic culture of forensic genetics, and its ethos (its diversity requires a shared sense of spirit). As such, we have emphasised this point within the introduction. We appreciate the call to define the community more clearly, and we do so in the introduction. An important point to make is that forensic genetics has a strong research community that also draws from and works with population genetics and specific forms of clinical genetics. Researchers tend to collect and/or use population data, and as such Cole’s analysis of social attributes only speaks to aspects of case work here, not to basic science. We make this point at the end of section 3.

In terms of the structure and organization of the text, it would be useful to the reader if the definition of each of the concepts - integrity, trust, and effectiveness - appeared immediately, in a very abbreviated format, at the beginning of the text, reserving the detailed description for the different sections that make up this article.

  • Response: Thank you, we have followed this suggestion early on in the introduction, and again as a reminder in the conclusion.

Finally, the authors' approach of 'trust' seems reductive to me, and I wonder what the real difference is in relation to other texts (an excellent example of this is the article “Value beyond utility: Let’s R.U.L.E.” (Wienroth, 2020; Journal of Responsible Innovation). I would like to see a more elaborated notion of trust, that also addresses “trustworthiness” (of individuals and institutions) and problematizes traditional accounts of “public trust”. An potentially useful reading might be  the text by Kieran O’ Doherty (2022), “Trust, trustworthiness, and relationships: ontological reflections on public trust in science” (Journal of Responsible Innovation)

  • Response: Thank you for the literature recommendations. We recognise the reviewer’s concerns around accounts of ‘public trust’ as from the start we have included a strong element of critically discussing trustworthiness of scientists and science in our manuscript. In this revision, we have expanded our existing discussion to emphasise our arguments, as requested by the reviewer. Please see the revised section on trustworthiness and trust.

Reviewer 2 Report

An excellent paper which will make an important contribution to the evolving field of forensic genetics. The paper is exceptionally well-written, referenced extensively and makes a solid contribution to the debate around use of this technology.

Only very minor comments:

- Page 4, line 182 "forensic imaginary" - I don't think this is the correct term. Do you mean the "forensic imagination"?

- Page 7, lines 267-273. I wasn't entirely taken by the GMO example. Is there a better example that can be used?  If not, then I would leave it as is. But it takes the reader out of the forensic genetics context, and seemed a little out of place in the paper.

- Page 11, line 491 - "as well as stakeholders".  It may be worthwhile being more specific around the types of stakeholders referred to here.

A pleasure to read - look forward to seeing this paper in print!

Author Response

Dear Dr. Turchi, dear Ms. Long,

We are grateful to the reviewers’ time for commenting on our manuscript, and we appreciate the opportunity to respond to their comments. We are particularly delighted by Reviewer 2’s comments: “An excellent paper which will make an important contribution to the evolving field of forensic genetics. The paper is exceptionally well-written, referenced extensively and makes a solid contribution to the debate around use of this technology. […] A pleasure to read - look forward to seeing this paper in print!”

We have considered all comments carefully, please find our responses below. We have tracked all text changes in the attached revised manuscript for your ease of access (and we have also attached a clean file with all changes accepted). NB: We spotted a doubling of a reference (previously as endpoints 59 and 63) and made the necessary changes to the order of endpoint references (from endpoint 63 onwards) which we have not tracked in the revised manuscript. Since we are working with a document provided by the journal now, any new / added references are numbered from the last reference endpoint onwards, irrespective of their place in text.

All the best,

The authors.

Reviewer 2

R2: An excellent paper which will make an important contribution to the evolving field of forensic genetics. The paper is exceptionally well-written, referenced extensively and makes a solid contribution to the debate around use of this technology. […] A pleasure to read - look forward to seeing this paper in print!

  • Response: Many thanks for these very encouraging words.

R2: Only very minor comments:

Page 4, line 182 "forensic imaginary" - I don't think this is the correct term. Do you mean the "forensic imagination"?

  • Response: Thank you, in this context the form of ‘imagination’ is indeed more appropriate, and we have made the change.

R2:

Page 7, lines 267-273. I wasn't entirely taken by the GMO example. Is there a better example that can be used?  If not, then I would leave it as is. But it takes the reader out of the forensic genetics context, and seemed a little out of place in the paper.

  • Response: This is a fair point, and we have changed the example to reference the discussion of forensic genetics methodology in US and UK courts from the late 1980s until the early 2000s, which required harmonisation and validation processes to take place in order to establish trustworthiness with courts.

R2:

Page 11, line 491 - "as well as stakeholders".  It may be worthwhile being more specific around the types of stakeholders referred to here.

  • Response: We agree and have made the following changes, deleting the term ‘stakeholders’ in this instance and replacing it with the following more encompassing wording: “and how to approach forensic testing and its communication to investigators and the judiciary as well as publics (e.g., local communities, victim groups, civil liberties organisations, policy-makers).”

Round 2

Reviewer 1 Report

The author(s) adressed very all the comments and suggestions. The paper is of high quality, and it is suitable for publication.

Author Response

Dear editors, dear reviewer,

Many thanks for reading our manuscript, we are glad to hear that our revision has addressed all your questions.

All the best,

The authors.